# Detecting Earthquake-Related Anomalies of a Borehole Strain Network Based on Multi-Channel Singular Spectrum Analysis

**DOI:** 10.3390/e22101086

**Published:** 2020-09-27

**Authors:** Zining Yu, Katsumi Hattori, Kaiguang Zhu, Chengquan Chi, Mengxuan Fan, Xiaodan He

**Affiliations:** 1Key Laboratory of Geo-Exploration Instrumentation, Ministry of Education, Jilin University, Changchun 130061, China; yuzn18@mails.jlu.edu.cn (Z.Y.); chicq15@mails.jlu.edu.cn (C.C.); mxfan19@mails.jlu.edu.cn (M.F.); hexd19@mails.jlu.edu.cn (X.H.); 2The College of Instrumentation and Electrical Engineering, Jilin University, Changchun 130061, China; 3Graduate School of Science, Chiba University, Inage, Chiba 263-8522, Japan; hattori@earth.s.chiba-u.ac.jp; 4Center for Environmental Remote Sensing, Chiba University, Inage, Chiba 263-8522, Japan

**Keywords:** borehole strain network, Multi-channel Singular Spectrum Analysis, network anomalies, self-organizing pre-earthquake phenomena

## Abstract

To investigate the nonlinear spatio-temporal behavior of earthquakes, a complex network has been built using borehole strain data from the southwestern endpoint of the Longmenshan fault zone, Sichuan-Yunnan region of China, and the topological structural properties of the network have been investigated based on data from 2011–2014. Herein, six observation sites were defined as nodes and their edges as the connections between them. We introduced Multi-channel Singular Spectrum Analysis (MSSA) to analyze periodic oscillations, earthquake-related strain, and noise in multi-site observations, and then defined the edges of the network by calculating the correlations between sites. The results of the daily degree centrality of the borehole strain network indicated that the strain network anomalies were correlatable with local seismicity associate with the earthquake energy in the strain network. Further investigation showed that strain network anomalies were more likely to appear before major earthquakes rather than after them, particularly within 30 days before an event. Anomaly acceleration rates were also found to be related to earthquake energy. This study has revealed the self-organizing pre-earthquake phenomena and verified the construction of borehole networks is a powerful tool for providing information on earthquake precursors and the dynamics of complex fault systems.

## 1. Introduction

Strain measurements play an essential role in analyzing geodynamic processes, such as those used to study slow earthquakes [1,2], volcanic activities [3], and preparation processes that occur prior to earthquakes [4,5,6,7]. In an effort to monitor the crustal activities associated with earthquakes, numerous geodetic monitoring systems have been established around the world, which provide an opportunity to investigate complex tectonic structures [8,9,10,11], short- and medium-term earthquake prediction [12,13,14], etc. The development of such monitoring systems has facilitated intensive studies on crustal deformations associated with earthquake preparation, their occurrence, and post-earthquake phases [9,15,16,17,18], such as those from the 2011 Tohoku earthquake in Japan [19,20,21] and the 2008 Wenchuan earthquake in China [22,23]. However, these monitoring systems currently consist of multiple sensors scattered across various and complex geographical locations. Consequently, understanding the interactions and relationships among observations becomes highly challenging.

Earthquake processes have complex spatial and temporal distribution characteristics, and the propagating environment of an earthquake is composed of a complex mechanical system within the crust. A topological network is an abstraction of a large number of real complex systems, expressing concealed interactions or relationships within a complex system. Abe and Suzuki, et al. pioneered complex network analysis in seismology to study seismicity as a spatiotemporal complex system, providing greater insights into seismicity patterns [24]. From this work, seismic catalogues have been converted into a complex network based on a multigraph, which is defined as an earthquake network. By computing the topological and dynamic characteristics of these networks, earthquake networks have revealed properties, i.e., small-world [25,26], scale-free [27,28] and hierarchical organizations [29]. In addition, the evolution of network structures indicates there have been precursory seismicity patterns before the occurrence of several earthquakes [30,31]. Therefore, we have attempted to use complex networks to investigate strain observations as a complex system in China, using the multiple YRY-4 borehole strainmeters that have been installed in the Sichuan-Yunnan region since 2008. YRY-4, as a four-gauge borehole strainmeters (FGBS), manufactured by [32], has been developed by the China Earthquake Administration as a means of updating the previous network. It has four gauges arranged at $45^{\circ }$ intervals in a cylindrical case, which measure the changes of a diameter in the corresponding azimuths. YRY-4 strainmeters have high observation accuracy over months to hours, and have been providing high-quality strain data for more than 10 years.

With regard to the observed geophysical time series, breakthroughs in technology used for sampling and data storing have advanced geodetic monitoring systems, enabling them to record data with larger frequency bands. Following this, the data decomposition method has been widely applied in feature extraction and target information separation, such as in Principal Component Analysis (PCA) [33,34] and Singular Spectral Analysis (SSA) [35,36]. By decomposing the observed geophysical time series, such as electromagnetic data and water level, researchers have investigated the eigenvalues or eigencomponents corresponding to seismic activities. This has revealed that the anomalies in time series are likely to be correlated with strong earthquakes [37,38]. However, PCA focuses on reducing the dimensionality of multivariate data sets, whereas observations are independent. In contrast, SSA considers the dependence of the observations for one-dimensional time series. Therefore, we used Multi-channel Singular Spectrum Analysis (MSSA) in this study to investigate the anomalous borehole strain changes. MSSA is particularly suitable for the multi-scale decomposition of spatio-temporal field data and for the analysis of spatio-temporal characteristics, which can be regarded as an extension of PCA and SSA. This technique is widely used in meteorology [39,40], oceanography [41], and other disciplines. MSSA can identify as much of the physical essence information as possible from the data and manage multivariate time-series by estimating periodic components and trends, and reducing noise [42,43,44].

To study the internal connection of the underground strain and reveal nonlinear spatiotemporal structures prior to larger earthquakes, it is necessary to adopt a macroscopic networked analysis method instead of working only with local observations. Therefore, in this study, we developed a complex network of multi-site borehole strain observations to provide new insights into the interactions of the strain field surrounding the southwestern endpoint of the Longmenshan fault zone and their relationship with local seismicity. Observation sites were defined as nodes in the network. Then, we identified the possible components related to earthquake activity using MSSA. By calculating the correlations between the selected components of each node, strongly correlated components were set to the edges of the network. To quantify the evolution of the borehole strain network structure, we monitored and computed daily network properties. Finally, we extracted the network anomalies and further discussed the relationship between these network anomalies and earthquakes within time and space.

## 2. Borehole Strain Observations and Studied Earthquakes

Multiple YRY-4 borehole strainmeters have been installed in southwestern China since 2008, aimed at monitoring the crustal activities associated with earthquakes near the Sichuan-Yunnan region. Among these monitors, six observation sites, Guza (GZ), Xiaomiao (XM), Zhaotong (ZT), Renhe (RH), Yongsheng (YS), and Tengchong (TC) provide a good opportunity to study the precursory information of earthquakes; all these have high quality data with long-term continuity. Figure 1 shows the locations of these six sites and the tectonic structure of the Sichuan-Yunnan region; further details are listed in Table 1. In this study, we used the borehole strain data collected from 2010 to 2014. These strainmeters sample every minute, and are capable of resolving strain changes of less than one-billionth. YRY-4 borehole strainmeters have a self-consistency function which is used to test the quality and credibility of the observations, as they have four gauges arranged at $45^{\circ }$ intervals. This arrangement produces four observation values: *S${}_{i}$*, (i=1,2,3,4) [11]. The reported data are the areal strain, *S${}_{a}$*, i.e., $S_{a}=\left(S_{1}+S_{2}+S_{3}+S_{4}\right)/2$, which describes the subsurface strain state [11].

Earthquakes occur when energy stored in elastically strained rocks is suddenly released. Local earthquake energy includes kinetic and strain energy carried by seismic waves, both of which depend on the magnitude of the earthquake and the hypocenter distance [35]. Therefore, in this study, we employed the $E_{s}$ parameter which considers both factors, in order to identify earthquake events. We selected shallow earthquakes (with depth of 0–60 km and magnitude greater than 1 (Ms≥1)) that occurred during 2011–2014 in the study region. The $E_{s}$ index is the daily sum of the local earthquake energy associate with strain network, $E_{s}^{\prime }$, whereas $E_{s}*$ is the energy of a local earthquake [35,45]. $E_{s}$ index is defined by the following equation:(1)$$E_{s}=\sum_{1day}E_{s}^{\prime }$$
(2)$$E_{s}^{\prime }=\sqrt[{N}]{\prod_{i=1}^{N}E_{s}*}$$
(3)$$E_{s}*=\frac{10^{4.8+1.5M}}{r_{i}^{2}}$$
where *M* and $r_{i}$ are the magnitude of the earthquake and the hypocenter distance (km) of each site, respectively, and *N* is the number of sites. The unit of $E_{s}$ index is J/km${}^{2}$, and is omitted in the following text for simplicity. To study the relationship between local seismicity and the borehole strain network, we set a study region to include the largest polygon formed by the sites and the area within 100 km from each site. Moreover, we adopted the following criterion: an earthquake event is considered to have occurred if the $E_{s}$ index on a given day exceeds $10^{7}$. Thus, there were 13 major earthquake events that counted within the strain network region. Details of these earthquake events are given in Figure 2a and Table 2, and the daily $E_{s}$ index is shown in Figure 2b.

## 3. Data Processing

### 3.1. Building a Borehole Strain Network

From the perspective of statistical physics, a network is a system that contains individuals and their interactions [46]. A typical network is composed of many nodes and some edges connecting the nodes. Nodes are used to represent different individuals in the real system, and edges are used to represent the relationships between individuals.

A borehole strain network is defined by a graph G=(V,E) where *V* denotes the set of nodes, and *E* denotes the set of edges between them. The nodes are defined as borehole strain observation sites; $V=v_{1},v_{2},\ldots ,v_{N}$, where *N* is the number of sites. The connections are defined by the significant correlation between the sequences of the two sites; this is detailed in Section 3.2. For simple edges, a two-dimensional matrix called an adjacency matrix was used; $A=a_{ij},i,j\in V$. If nodes *i* and *j* are connected, the corresponding element $a_{ij}$ in the matrix is set to 1; if they are not, it is set to 0. For time series of two nodes, $P=P_{i}$ and $Q=Q_{j}$, i,j∈V, $p_{t}$ and $q_{t}$ are samples of $P_{i}$ and $Q_{i}$, respectively. The Pearson’s correlation coefficient is defined as
(4)$$R_{PQ}=\frac{\sum_{t=1}^{n}\left(p_{t}-\bar{p}\right)\left(q_{t}-\bar{q}\right)}{\sqrt{\sum_{t=1}^{n}{\left(p_{t}-\bar{p}\right)}^{2}\sum_{t=1}^{n}{\left(q_{t}-\bar{q}\right)}^{2}}}$$
$\bar{p}$ and $\bar{q}$ are the means of *P* and *Q*, respectively. The adjacency matrix $A=a_{ij},i,j\in V$ acquires the components that were assigned the value of 1 if the absolute value of $R_{PQ}$ was greater than 0.8 and 0 is less than 0.8. Only the undirected graph ($a_{ij}=a_{ji}$) was considered in this study.

To monitor the borehole strain network structure, the network properties can be computed based on the adjacency matrix A. The degree $k_{i}$ of a node in a borehole strain network refers to the total number of edges connected to it:(5)$$k_{i}=\sum_{j=1}^{N}a_{ij}$$

Then, the daily degree centrality can be computed to evaluate the changes of the strain network connection:(6)$$\bar{k}=\frac{1}{N}\sum_{i=1}^{N}k_{i}$$

The greater the degree centrality, the stronger the network connectivity; otherwise, the nodes are considered to be weakly connected. The ‘degree’ mentioned in subsequent sections of this paper refers to the degree centrality.

### 3.2. Connecting Edges of the Borehole Strain Network Based on MSSA

It is challenging to find correlations between their raw time series for the strain sequences for different geographic locations. Therefore, we used MSSA to capture the spatio-temporal correlation behavior. MSSA stems from SSA and has similar steps [47]. There are observation sequences of *N* stations in the study area, $X_{l,t}$, for l=1,2,…N and t=1,2,…n. MSSA forms the trajectory matrix *Y* with a time lag *m* (a lag-window) to be analyzed:(7)$$Y=\left[\begin{array}{cccc} X_{1,1} & X_{1,2} & \cdots & X_{1,n-m+1}\\ X_{1,2} & X_{1,3} & \cdots & X_{1,n-m+2}\\ \vdots & \vdots & \ddots & \vdots \\ X_{1,m} & X_{1,m+1} & \cdots & X_{1,n}\\ X_{2,1} & X_{2,2} & \cdots & X_{2,n-m+1}\\ \vdots & \vdots & \ddots & \vdots \\ X_{N,m} & X_{N,m+1} & \cdots & X_{N,n} \end{array}\right]$$
where Y is a Nm×(n−m+1) matrix. Generally, the larger *m* is, the better the extraction of long-period oscillations. Smaller values of *m* are better for extracting high-frequency components. In our study, we set values of 4, 1440, and 6 for *m*, *n* and *N*, respectively.

Secondly, Y can be decomposed by the singular value decomposition (SVD), i.e., $Y={\mathrm{USV}}^{T}$, in which S is a diagonal matrix with *d* singular values $\left(\lambda_{k},k=1,2,\ldots d,d=Nm.\right)$ sorted in descending order. Each component $T_{k}$ corresponds to the matrix $T_{k}=\sqrt{\lambda }u_{k}v_{k}^{T}$ in which $u_{k}$ and $v_{k}$ are the left singular vector and the right singular vector for the singular value $\lambda_{i}$ is of the matrix Y. The reconstructed sequences (RCs) $Y_{l,t}^{k}$ can be obtained via the diagonal averaging of the matrix $T_{k}$,
(8)$$Y_{l,t}^{k}=\left\{\begin{array}{cc} \frac{1}{t}\sum_{i=1}^{t}T_{i,t-i+1}, & 1\leq t<m\\ \frac{1}{m}\sum_{i=1}^{m}T_{i,t-i+1}, & m\leq t<n-m+1\\ \frac{1}{n-t+1}\sum_{i=t-n+m}^{m}T_{i,t-i+1}, & n-m+1\leq t<n. \end{array}\right.$$

As borehole strain data include different signal types such as the response of the solid earth tide, air pressure, water level, tectonic changes and noise [48], in the decomposed component, it is expect to find the separated contribution of solid Earth tide, air pressure, and so on. As we know, the reconstructed components are ordered by decreasing contribution about the original time series. We found that the first 20, 15, and 8 components contributed 99.99%, 99.90%, and 99.00% of all the components, respectively. In this work, we not only want to find the main characteristics of the original data and eliminate the fairly small amount of noise, but also ensure that high-frequency signals related to earthquakes are included in the effective components. Therefore, the first 20 components were used using the following equation. They were then used to select the appropriate subset to reconstruct the strain data using the w-correlation method [49], which is as follows:(9)$$\rho_{i,j}^{w}=\frac{\left(Y^{\left(i\right)},Y^{\left(j\right)}\right)}{{∥Y^{\left(i\right)}∥}_{w}{∥Y^{\left(j\right)}∥}_{w}}$$
where ${∥Y^{\left(i\right)}∥}_{w}=\sqrt{(Y^{\left(i\right)},Y^{\left(i)\right)}}$ and $\left(Y^{\left(i\right)},Y^{\left(j\right)}\right)=\sum_{t=1}^{n}w_{t}y_{t}^{i}y_{t}^{j}$ in which $w_{t}=min\left(t,m,n-t\right)$. The large absolute ρ indicates that $Y^{\left(i\right)}$ and $Y^{\left(j\right)}$ can possibly be grouped together. Typically, the ρ of the first six RCs was very large; therefore, we considered that these corresponded to the different components of solid tides and to changes in low frequencies possibly related to the environment. These signals can be called periodic components. Therefore, we extracted the components with larger ρ values from the last 14 RCs for consideration as earthquake-related strain and denoted the rest of the RCs as being composed of noise. The sliding correlations between the extracted strain changes were then computed using Equation (Equation 4). The highest correlation coefficient of the day exceeding 0.8, implied an edge between the two sites. Figure 3 illustrates examples of the strain network results of an aseismic day and an anomaly day. We applied MSSA to decompose the raw observations from the six nodes; then, we grouped the RCs using the *w*-correlation method into periodic components, earthquake-related strain changes (only on anomaly days) and noise, and then demonstrated the corresponding networks. In fact, the anomalous day in Figure 3 was found to correspond to 13 April 2011, a day on which a magnitude 3.0 earthquake occurred at 5:12 (UTC+8), at $30.61^{\circ }$ N, $103.25^{\circ }$ E. This verified that the correlations of strain changes between the nodes were very strong and that the network was fully connected.

### 3.3. Definition of Borehole Strain Network Anomalies

After the edges were thus defined, a borehole strain network was established. For this study, a temporal window of one day was set to infer the strain changes of the study region by observing the changes in the daily network structure. The daily degree of the six-node network in the study region is detailed in Figure 4a. To detect anomalies through an appropriate threshold, we checked the empirical probability distribution of the three-year average degrees in Figure 4b, and found the distribution was significantly skewed and extreme.

As we know, when the Earth’s crust is deformed, such as during earthquakes or underground movements, the strainmeters in the affected space record these changes. According to the definitions of edges above, this causes the strain network to be either fully connected or strongly connected. Thus, there are several extreme values within the degree distributions. In Figure 4b, we found that there was a second peak at approximately 3.33, and these large values, which corresponded to strongly connected networks, may be related to anomalous underground activities. Therefore, we considered that a threshold greater than this value would be reasonable. In this study, we set the threshold to 4, i.e., 80% or more of the fully connected networks were regarded as network anomalies.

## 4. Results

### 4.1. Comparison of Borehole Strain Network Anomalies and Local Seismicity

According to previous studies [9,50,51], borehole strainmeters are capable of recording subsurface seismic strain steps with a high sampling rate. Thus, in our strain network detection, the recorded co-seismic strain changes may be one of the main sources for strongly connected networks. Therefore, we investigated the relationship between borehole strain networks and local seismic activities.

First, we calculated all the local earthquake energy in the study area from 2011 to 2014. We then analyzed the energy distribution as illustrated in Figure 5a, and found that the most probable local earthquake energy values were $10^{2}$ and $10^{3}$. As the energy increased, the number decreased exponentially. Then, we counted the distribution of degrees in each energy interval, as shown in Figure 5b. In each energy interval, the distribution of degrees was still skewed and extreme. In particular, degrees of 4 to 5 seemed to account for a certain amount. Following this, the weights of degree anomalies in each energy interval were computed as shown in Figure 5c. We found that the weight of degree anomalies increased as the local earthquake energy increased for energy values greater than $10^{5}$. This indicated that there was a correlation between strain network anomalies and earthquakes with energy greater than $10^{5}$. Therefore, we consider that the network anomalies can reflect the anomalous underground deformation, which is very important information when studying earthquake precursors. In addition, this also verified the threshold selected had the ability to distinguish network anomalies.

However, some degree anomalies also occurred when the earthquake energy was less than $10^{5}$, implying that there were some anomalies for which network enhancements might not have been related to local seismic activities. On the one hand, the strain network may have monitored some small earthquakes. On the other hand, the strain network may have recorded other sources of strain.

### 4.2. Network Anomalies Associated with Major Earthquakes

The internal physical properties of earthquake processes are complex. However, some earthquakes with larger energy have been reported to generate various patterns before their occurrences [21,52,53], including foreshocks [54,55], preseismic quiescence [56,57], precursory spatiotemporal clusters [33,58], and earthquake critical phenomena [53,59]. It is currently impractical to expect that preseismic anomalies would show significant event-to-event correspondence with earthquakes. In Section 4.1, we analyzed the statistical relationship between local seismicity and strain network anomalies. In order to further verify their temporal correlation, we performed a case study on the network anomalies before and after each major earthquake.

First, we selected the earthquake event when the $E_{s}$ index in that day exceeded $10^{7}$. During 2011–2014, there were 13 earthquake events within the study area adopted, as introduced in Section 2. Second, for each earthquake event, we created a window of network anomalies for 90 days before and after its occurrence day. Next, we investigated the cumulative counts of network anomalies as previously defined and repeated this procedure for all 13 earthquake events selected. To highlight the increased and clusters of anomalies from the cumulative anomalies associated with the major earthquakes, the Sigmoid function was used to fit the cumulative anomalies around their occurrence days [14]. The function can be expressed as follows: (10)$$y=A2+\frac{\left(A1-A2\right)}{\left(1+e^{\frac{x-x0}{dx}}\right)},$$
where A1, A2, x0, and dx are the asymptotic lower limit, asymptotic upper limit, inflection point, and time constant, respectively. This function is characterized by two different behaviors with opposite concavities before and after a center point (x0); dx can be seen as a reflection of the acceleration rate of the function. The smaller the dx value, the faster the function grows, and vice versa. These parameters have been widely used to describe seismic anomalies [13]. The results are presented in Figure 6. The fitting parameters are provided in the right corner of each graph. Adj R-square is goodness of fit, the closer to 1, the better is. Almost all the Adj R-square of fitting exceeded 0.97, indicating the fitting results were very convincing.

In Figure 6, the cumulative number on any given day indicates the sum of strain network anomaly counts before that day. On the one hand, if the earthquake events and anomalies have no correlation, the accumulation of random anomalies would be expected to increase linearly. On the other hand, a marked increase in the slope of the accumulation curve prior to an earthquake indicates that there is a possible relationship between the anomalies and external strain sources. Figure 6 shows significant surges appear prior to most major earthquakes such as EQ3–EQ13, and the time at which the network anomalies began to appear varied from 20 to 60 days before the event. In addition, the network anomalies tended to be stable after most of the earthquakes. It was noted that, for EQ2, which occurred between EQ1 and EQ3, the network anomalies abruptly enhanced after the EQ1 and lasted for nearly 40 days, so the network has not changed significantly for EQ2. Furthermore, EQ1–EQ4 and EQ7 occurred near the TC node, which was relatively distant from the center of the strain network compared to some of the other earthquake events. This difference in geographic location could be one of the reasons for the difference in the cumulative results from the other major earthquakes. We didn’t show EQ12 because it was the second day of EQ11, and it was almost the same as EQ11. However, the vast majority of the results strongly suggested that there was a correlation between the major earthquake events and strain network anomalies.

To further investigate the characteristics of network enhancement before and after the earthquakes, we calculated the slopes of the fitted anomaly accumulations as temporal cumulative rates, as shown in Figure 7a. In Figure 7a, of the 13 earthquakes, 11 of the cumulative rates reached their maximums before the earthquake. Furthermore, all these maximum values occurred within 30 days before the event.

Next, we studied the relationship between these anomaly accumulations before the earthquake and the corresponding earthquake energy. However, the cumulative rate of each earthquake could not be compared directly because the counts and the length of time for each fitting were different. We thus introduced dx in the Sigmoidal function as it can reveal the accumulation acceleration to help us to evaluate the network anomalies and earthquake energy. Figure 7b shows the relationship between the energy and the cumulative acceleration rate of nine earthquakes. We excluded EQ1–EQ2 because they did not accelerate significantly before the earthquake. We also excluded EQ12 because it is the second day of EQ11 with many aftershocks, so its accumulation could not be isolated. The colors indicate corresponding earthquakes. In Figure 7b, the local earthquake energy and cumulative acceleration rates exhibits a linear relationship. The greater the energy, the smaller the dx, indicating the faster the accumulation. It is worth noting that EQ10 took place three days before EQ11; thus, the relationship between its pre-earthquake anomalies and EQ11 is indiscernible. If EQ10 is removed, the goodness of the linear fit increases from 56.83% to 74.56%. Figure 7a indicates that the strain networks are more likely to enhance before rather than after major earthquakes ($E_{s}>10^{7}$). Combining this with the relationship shown in Figure 7b, we considered that the strain network anomalies may be associated with the major earthquakes.

## 5. Discussion

During the last decade, considerable progress has been made towards understanding preseismic processes. In this study, we attempted to understand whether the properties of borehole strain data are associated with earthquake activities. We found that there would be an enhanced connection of monitoring network before major earthquakes. First, the enhanced connection of the strain network was considered to indicate the self-organization of the system and the clustering of strain anomalies. A large number of precursory studies have demonstrated orderly and self-organizing pre-earthquake phenomena. For example, successive earthquakes were studied as a spatiotemporal complex system through the evolution of network measures, which revealed the underlying organization principles of earthquake networks. They found that the network measures exhibited an abrupt jump shortly before the main shocks [31]. Precursory patterns observed in the seismicity a few days before the mainshock (that may even lead to earthquake prediction) have also been revealed by many other studies [53,60,61]. Sarlis, et al. investigated the behavior of seismicity after the observation of the Seismic Electric Signal activity using natural time analysis and revealed the fluctuations of the order parameter of seismicity exhibited a minimum a few days before a major earthquake [53]. In addition, this study recently has been verified to show statistical significance by means of the receiver operating characteristics (ROC) technique by focusing on the area under the ROC curve (AUC) [62] and event coincidence analysis [59]. In addition, entropy and negentropy, which serve as measures of the unknown external energy inflow and outflow the seismic system, also exhibited self-organization properties for catastrophic events [14,58,63]. Karamanos, et al. quantified and visualized temporal changes in complexity using approximate entropy [64,65], and they stated that a significant decrease in complexity and accession at the tail of pre-seismic electromagnetic emissions can be a diagnostic tool for characterizing an impending earthquake. The fault network inside Mainland China can be considered as a big seismogenic system [66]. In some subsystems (fault blocks, fault activities, etc.), stress accumulation and crustal movement will occur, and multiple ‘potential sources’ will be formed. The spatiotemporal evolution of the interaction between various subsystems and ‘potential sources’ is a self-organization process, which is ‘steady state-deviation from steady state-instability-new steady state’. Whatever the expression of the system and the change in the data, before reaching the new steady state, there will be a reduction phenomena in entropy, dimensionality and disorderly [66]. Here, the earthquake precursor is integral dynamic behavior during the system evolution process and also the tectonic micro-dynamic behavior during the self-organization evolution of the seismogenic system. In terms of borehole strain, network enhancements prior to major earthquakes may reflect underground crustal movements have begun to deviate from the steady state and show self-organization, while the recovery of the network after an earthquake could represent the reaching of a new steady state.

Second, there was a correlation between strain network anomalies and earthquakes with larger energy as shown in Figure 5. In order to confirm whether the detected network anomalies are due to the foreshock activities or underground strain changes, we further computed the cumulative $E_{s}$ values using the same procedure as above. We set $E_{s}$ threshold as $10^{5}$. If the $E_{s}$ value greater than this threshold, we counted one for the corresponding day, then added up the counts in the same window (90 days before and after the event day). The accumulation of the $E_{s}$ values greater than $10^{5}$ for each event are shown in Figure 8. Compared with strain network anomalies for the 13 major earthquakes, the results of $E_{s}$ values didn’t show the pre-earthquake enhancements. The $E_{s}$ accumulation of some earthquakes have similar characteristics, thus we laid them in four sub-graphs (EQ12 was also omitted). In the 40 to 20 days before the EQ1 and EQ2, local seismicity increased, but, in the strain network, the anomalies enhanced after the earthquake. In the first half year of 2011, earthquakes were active near the TC station. However, the station is far away from the central area of the strain network, so it was reasonable that the results were inconsistent. Before and after the EQ3–EQ4 and EQ13, there were no significant changes, and the linear fits also showed that these major earthquakes had little effect on local seismicity. There were also no increases in seismicity prior to the EQ5–EQ7 and EQ8–EQ11. On the contrary, possibly due to the aftershocks of EQ5–EQ7 and EQ10–EQ11, the accumulations of these earthquakes had several significant overlaps. The results showed that the foreshock activities were different from the detected changes of the strain network before major earthquake events. Therefore, the strain network anomalies before major earthquakes ($Es>10^{7}$) were more likely to be related to the underground strain changes than their foreshocks.

Third, to investigate whether distant earthquakes may generate the network anomalies, we analyzed the earthquake energy as Equation (Equation 1) for a larger region, i.e., $24^{\circ }$–$31^{\circ }$ N and $97^{\circ }$–$107^{\circ }$ E. There were five additional earthquakes with energy values greater than $10^{7}$, as shown in Table 3 and Figure 2. We also studied the cumulative anomalies, as shown in Section 4.2 for each earthquake in Figure 9a. $EQ_{b}-EQ_{d}$ were very close, so they were put on one graph, and $EQ_{c}$ was omitted for a clearer view. However, only before $EQ_{e}$ was there an accelerated increase in the network anomaly accumulation. $EQ_{a}-EQ_{d}$ exhibited few anomalies before the earthquake, and the anomalies after the earthquakes are not likely to be as closely related. Network anomalies, in general, can be generated by many factors, and the local earthquake is just a possible one. Thus, we also investigated their relationships with the local seismicity as shown in Figure 9b. The results showed that the accumulations of $E_{s}$ values ($E_{s}>10^{5}$) for $EQ_{b}-EQ_{d}$ were influenced by the aftershocks of $EQ_{b}$. However, for $EQ_{a}$ and $EQ_{e}$, the accumulation of network anomalies and $E_{s}$ values ($E_{s}>10^{5}$) were similar. In particular, 30 to 10 days before the $EQ_{e}$, there was a similar acceleration. In this way, the accelerated increase before the event was possibly related with the foreshocks. Therefore, we found that the strain network anomalies were more sensitive with the closer earthquake events in the study region rather than distant events. These results also suggested that the epicenter distance should be considered when discussing the earthquake precursor anomalies.

We could neither expect that there would be anomalies before each earthquake, nor could we expect that each anomaly would correspond to an earthquake. Therefore, we may be able to improve the monitoring of the preparation stage of major earthquakes through interdisciplinary investigations and multiple observations [67]. The most remarkable cumulative rate and cumulative acceleration rate of EQ11 shown in Figure 7 refer to the 2013 Lushan earthquake, the largest earthquake in China after the 2008 Wenchuan earthquake. In the Sigmoidal fit for EQ11, the network anomalies from 16 to 20 April were the main reasons for the abrupt increase in the accumulation and the very small dx value. Many other studies have also focused on the short-term impending anomalies for the earthquake [68,69,70,71]. The investigation of the ionosphere VTEC anomalies using GPS data [68,72] and the geo-electric field variation [73] indicated that the anomalies occurred five days before the earthquake. These highly consistent results make the extracted strain anomalies more convincing. Therefore, we considered that the strain network anomalies from 16 April may be related to the Lushan earthquake.

## 6. Conclusions

In order to verify the existence of recognizable strain changes preceding major earthquakes, network analysis was conducted using borehole strain data from the southwestern endpoint of the Longmenshan fault zone, China, for the period 2011 to 2014. First, local earthquake energy $E_{s}$ corresponding to a multiple-site borehole site network was employed to select the earthquake events. Then, to complete the borehole strain network, six observation stations were selected as nodes, and the earthquake-related components were separated using MSSA to define edges. Finally, we computed the daily degree centrality to evaluate the changes of the strain network connection and selected over 80% of fully connected networks as anomalies, thereby indicating connections in the network enhancements. The results indicated that the borehole strain network anomalies and local seismicity ($E_{s}>10^{5}$) were correlated. Moreover, of the 13 major earthquakes ($E_{s}>10^{7}$), there were 11 acceleration increases in network anomalies before the events that we verified that they were likely not due to foreshock activity. Analysis of the anomaly cumulative rates indicated that the enhancements in the strain network were more likely to appear before rather than after major earthquakes, generally within 30 days before the events. Further investigation suggested that the local earthquake energy was related to cumulative anomaly acceleration rates. Therefore, we consider that network anomalies could be associated with major earthquakes and could reflect a self-organizing phenomenon of underground activities. Thus, strain network analysis has the potential to help understanding of the earthquake generation process. Furthermore, MSSA has been demonstrated to be a useful and effective tool. Future studies should consider strain data for a longer time and a broader region and focus on statistical analysis of borehole strain networks.

## Figures and Tables

**Figure 1 entropy-22-01086-f001:**
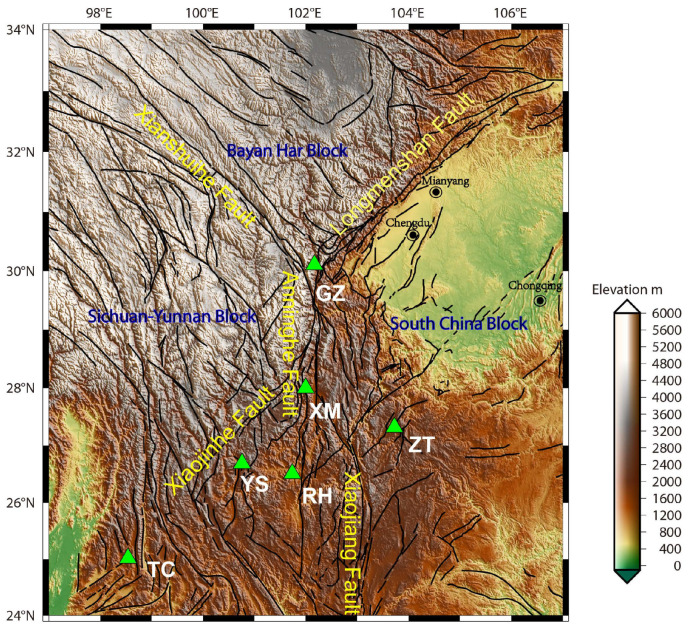
Map showing tectonic structure of the Sichuan-Yunnan region and the locations of the six borehole strain observation sites (green triangles). The black curves indicate the faults.

**Figure 2 entropy-22-01086-f002:**
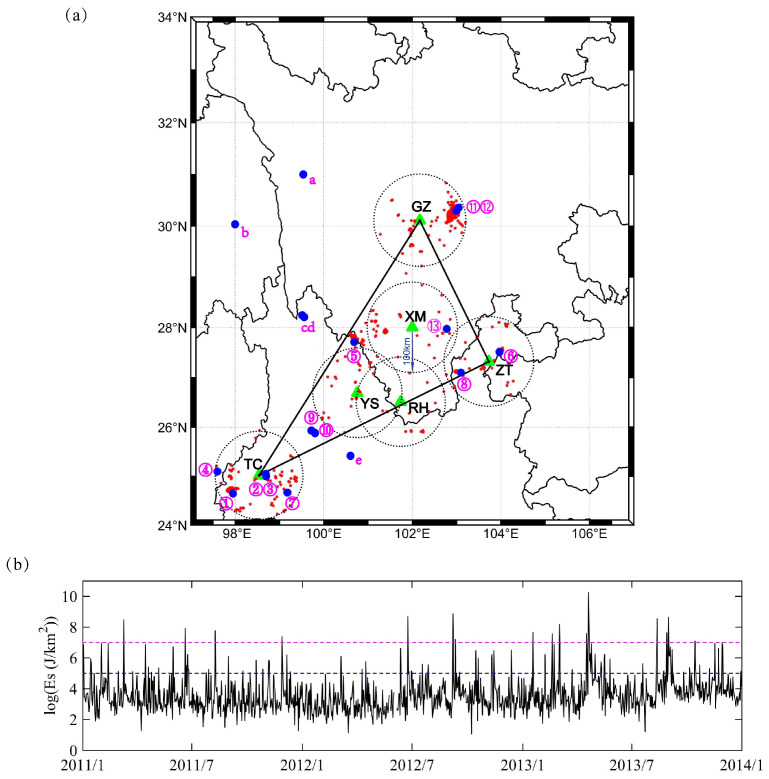
(**a**) the strain network region: the maximum observation network based on the site locations (black triangle) and the area within 100 km (black dotted circle) of each site. Blue dots numbered 1–13 show the spatial distributions of 13 major earthquakes with $E_{s}>10^{7}$ in the strain network region during 2011–2014. Red dots indicate earthquakes (Ms>2) in the region. Blue dots labelled a−e show five additional major earthquakes with $E_{s}>10^{7}$ in $24^{\circ }$–$31^{\circ }$ N and $97^{\circ }$–$107^{\circ }$ E. (**b**) The $E_{s}$ variations in the study region. The pink and blue lines are thresholds of $10^{5}$ and $10^{7}$ for $E_{s}$ values, respectively.

**Figure 3 entropy-22-01086-f003:**
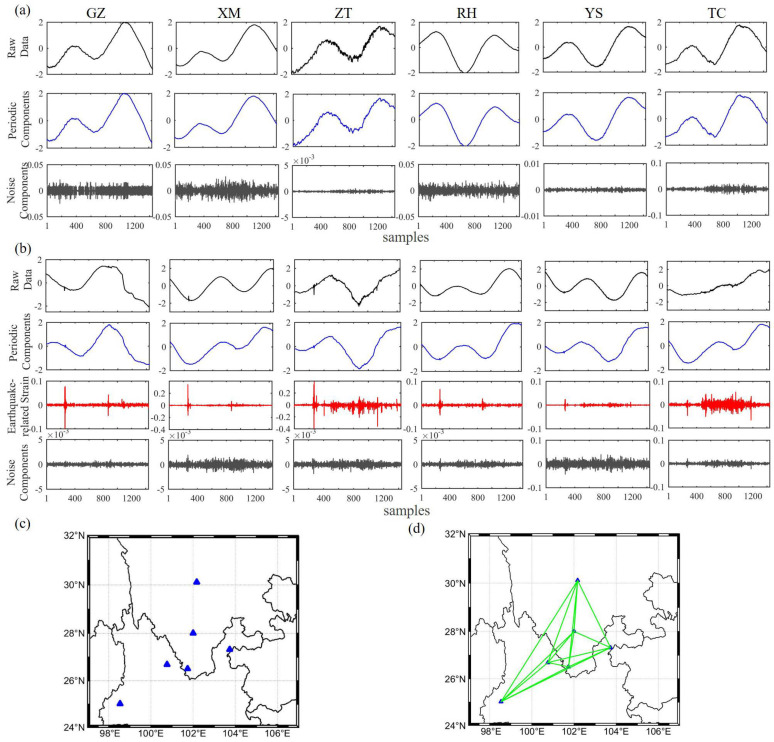
(**a**) an example of the MSSA results of an aseismic day. Black lines indicate the raw data of the six nodes, while the blue and grey lines represent the periodic components and noise components of the strain, respectively; they correspond to the six nodes after MSSA; (**b**) an example of the MSSA results of an anomaly day. Black lines indicate the raw data of six nodes; red lines are the grouped earthquake-related strain changes after MSSA, and blue and grey lines are the periodic components and noise components of the strain, respectively; (**c**) the borehole strain network corresponding to (**a**). There are no connected edges in the network for an aseismic day; (**d**) the borehole strain network corresponding to (**b**). This is a fully connected network.

**Figure 4 entropy-22-01086-f004:**
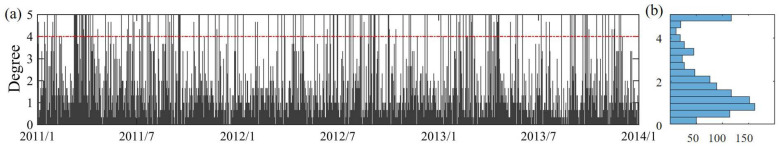
(**a**) the daily degree of the borehole strain network during 2011–2014. The horizontal red dotted line indicates 80% of the fully connected networks; (**b**) histogram of the three-year degrees.

**Figure 5 entropy-22-01086-f005:**
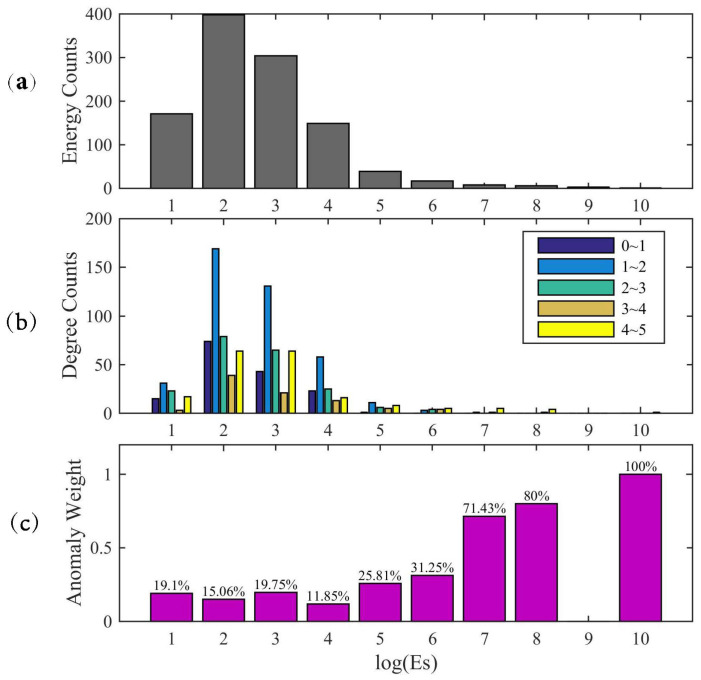
(**a**) local earthquake energy distributions at different energy intervals; (**b**) the degree distributions at different energy intervals; (**c**) the weights of degree anomalies at different energy intervals.

**Figure 6 entropy-22-01086-f006:**
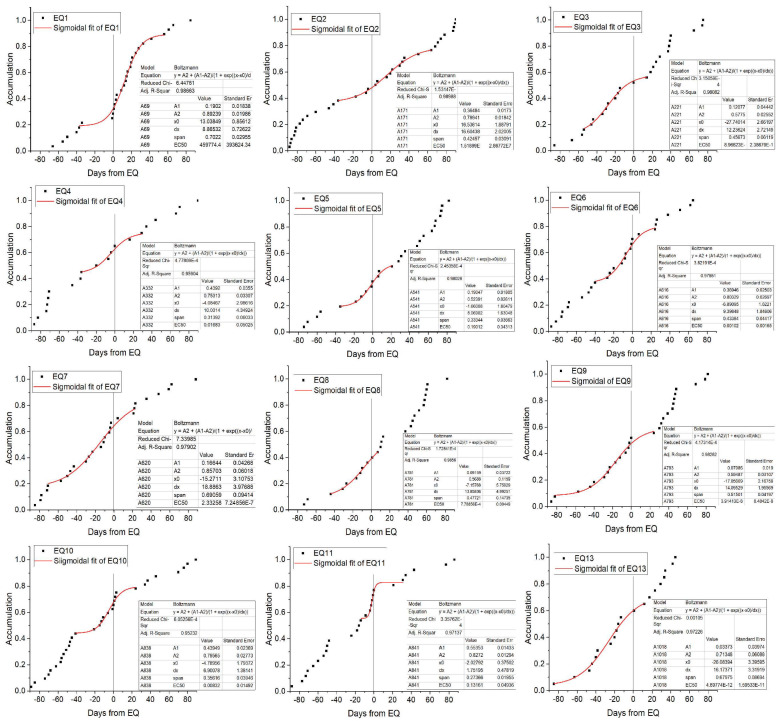
Accumulation results for borehole strain network anomalies of 13 major earthquakes. Black squares show the cumulative degree anomaly counts for 90 days before and after the earthquake day. Red lines show the Sigmoidal fits of the cumulative degree anomalies around the earthquakes. Vertical black lines (at the 0 day) indicate the day when $E_{s}$ is greater than $10^{7}$.

**Figure 7 entropy-22-01086-f007:**
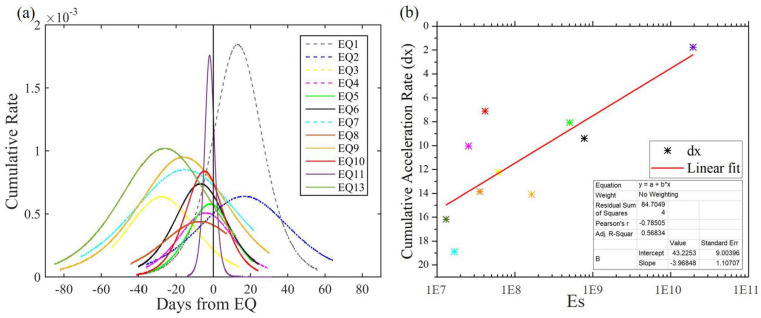
(**a**) cumulative rates of the degree anomalies before and after the 13 major earthquakes (the vertical black lines at the 0 day). The colors indicate corresponding earthquakes; (**b**) correlation between local earthquake energy and cumulative acceleration rate of EQ3–EQ13, excluding EQ12 as it is the second day of EQ11; it was omitted as its accumulation was considered to come mainly from EQ11. The colors indicate corresponding earthquakes.

**Figure 8 entropy-22-01086-f008:**
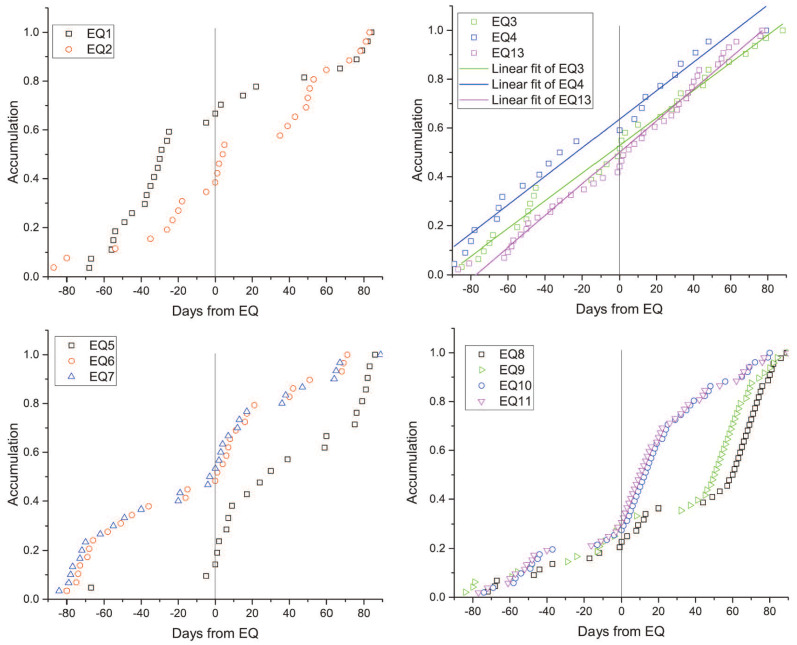
Accumulation results for $E_{s}$ values of the 13 major earthquakes. Hollow labels show the cumulative counts ($E_{s}>10^{5}$) for 90 days before and after the major earthquake day. Vertical black lines (at the 0 day) indicate the day when $E_{s}$ is greater than $10^{7}$.

**Figure 9 entropy-22-01086-f009:**
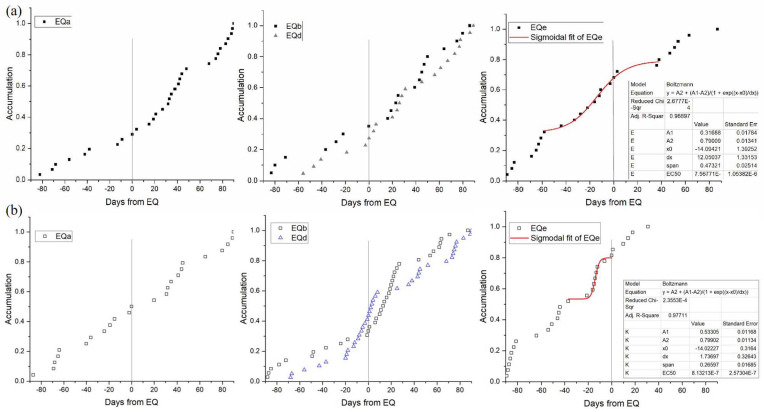
(**a**) accumulation results for borehole strain network anomalies of four additional earthquakes; (**b**) corresponding accumulation results for $E_{s}$ values. Scatters show the cumulative degree anomaly counts and $E_{s}$ value counts ($E_{s}>10^{5}$) for 90 days before and after the earthquake day, respectively. Red lines show the Sigmoidal fits of the cumulative degree anomalies around the earthquakes. Vertical black lines (at the 0 day) indicate the day when $E_{s}$ exceeded $10^{7}$. The table in the lower right corner provides the fitting parameters.

**Table 1 entropy-22-01086-t001:** List of Borehole Strain Stations Shown in Figure 1.

Station Name (Code)	Geographic Coordinates	Type of FGBSs	Sampling Rate
Guza (GZ)	$30.11^{\circ }$ N, $102.17^{\circ }$ E	YRY-4	1 min
Xiaomiao (XM)	$28.00^{\circ }$ N, $102.00^{\circ }$ E	YRY-4	1 min
Zhaotong (ZT)	$27.32^{\circ }$ N, $103.73^{\circ }$ E	YRY-4	1 min
Renhe (RH)	$26.51^{\circ }$ N, $101.74^{\circ }$ E	YRY-4	1 min
Yongsheng (YS)	$26.69^{\circ }$ N, $100.76^{\circ }$ E	YRY-4	1 min
Tengchong (TC)	$25.02^{\circ }$ N, $98.54^{\circ }$ E	YRY-4	1 min

**Table 2 entropy-22-01086-t002:** List of 13 major earthquakes with $E_{s}>10^{7}$ within the strain network region.

No.	Date	Longitude (${}^{\circ }$E)	Latitude (${}^{\circ }$N)	Depth (km)	Magnitude ($M_{s}$) ${}^{a}$	$E_{s}$ (×10^7^) ${}^{a}$
1	10 March 2011	97.95	24.64	10	5.9	31.20
2	20 June 2011	98.69	25.04	10	5.3	8.86
3	9 August 2011	98.70	25.00	11	5.2	6.11
4	28 November 2011	97.60	25.15	10	5.2	2.54
5	24 June 2012	100.69	27.71	11	5.7	50.56
6	7 September 2012	103.97	27.51	14	5.7	78.00
7	11 September 2012	99.18	24.67	14	4.9	1.67
8	19 February 2013	103.10	27.10	10	4.9	3.56
9	13 March 2013	99.72	25.93	9	5.5	16.41
10	17 April 2013	99.75	25.90	10	5.4	4.14
11	20 April 2013	102.99	30.30	17	7.0	1940.50
12	21 April 2013	103.05	30.36	27	5.0	11.49
13	14 October 2013	102.78	27.97	18	4.6	1.31

${}^{a}$Ms represents the greatest magnitude on a day and $E_{s}$ is the energy sum of the day.

**Table 3 entropy-22-01086-t003:** List of additional earthquakes with $E_{s}>10^{7}$ within a larger region.

No.	Date	Longitude (${}^{\circ }$ E)	Latitude (${}^{\circ }$ N)	Depth (km)	Magnitude ($M_{s}$)	$E_{s}\left(\times 10^{7}\right)$
a	18 January 2013	99.40	30.95	15	5.5	4.90
b	12 August 2013	97.96	30.04	15	6.1	37.08
c	28 August 2013	99.33	28.20	9	5.2	4.52
d	31 August 2013	99.35	28.15	10	5.9	46.98
e	28 November 2013	100.58	25.40	19	4.7	1.01

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
