# Peer review of "Detecting Earthquake-Related Anomalies of a Borehole Strain Network Based on Multi-Channel Singular Spectrum Analysis"

_entropy, 2020, doi:10.3390/e22101086_

Round 1

Reviewer 1 Report

Revision of “Detecting earthquake-related anomalies of a borehole strain network based on multi-channel singular spectrum analysis” by Yu et al., Entropy

The paper presents a very interesting analysis of from a strain network composed of 6 stations. The authors analysed the system as a “mathematical network” searching for the possibility that the network shows a self-organizing features before major events in the region (Sichuan, China). Several case studies have been analysed by the authors in this paper including a significant earthquake: M7.0 Lushan 2013. The data are well processed and deeply analysed, so the paper is sure valuable to be published.

I have a doubt if suggest a major or minor revision because there is a delicate point on figure 6 and in general, if the pre-event identified acceleration is a strain pre-cursor or simply due to eventual foreshock activity, but anyway the work is valuable, but I would revise again the paper after further analysis of seismicity in the region (cumulate of the ES for example in the 12+4=16 discussed events).

Some general comments:

  • I suggest to add some details about the YRY-4 strainmeters, as the instrument is not described in the paper.
  • In Figure 6 I strongly suggest to add the cumulate of the daily released energy or the Es. As you show on Figure 5 there is a correlation between the high degree, i.e. the anomalies and log(Es), so I don’t want that the shape of fig. 6 is due only to foreshock activity, in this case, the detected change of the strain before (some?) of the big earthquakes is due only to seismic activity and it is not a precursor.
    The cumulate of Es for each seismic event (excluding EQ-12) can be over-plotted or presented in another picture in the best way the authors think.
  • In Acknowledgment is reported the earthquake catalogue source. I suggest reporting also in the test specifying which magnitude threshold has been used (if any) and preferably reporting also some discussion of magnitude distribution of the analysed / catalogue events in the region (Gutenberg-Richter diagram for example or any other).
  • If the authors agree you can add a citation that is an updated paper of the [65] published exactly in this Journal:
    De Santis et al., (2019) “Geosystemics View of Earthquakes.” Entropy, 21(4), 412; https://doi.org/10.3390/e21040412

Specific points:

  • Page 1 line 2/3. I think it’s better to specify in addition witch region of China: Sichuan-Yunnan, right?
  • Page 1 line 5. Multi-channel Singular Spectrum Analysis. I think with uppercase of the first letters of the acronym it si more clear (also for the other one following in the paper).
  • Page 1 line 9. The quantity ES needs to be introduced in the abstract if used. Furthermore is it Joule or √(J)/m, please add the unit of measure if any.
  • Page 2 line 40. Even if the sentence is right I suggest to replace one of the two “complex networks” and “network structures” to avoid the repetition and make the sentence more clear!
  • Page 2 line 66. “the components of earthquake-related behavior”. Is it: “the possible components related to earthquake activity”?
  • Page 3 line 82. Please, provide the equation for Sa
  • Page 3 last lines (number of line lost). “Es’ is the energy”. From the equation 2 I think the energy is the numerator inside the square-root, so not Es’, could you please explain better or revise this?
    Please, note that this possible problem is also I table 2 and other parts as line 89…
  • Page 4 line 90. “occurred” I think is not the best word as also the lower energy earthquakes occur, you can substitute by something like that are significative for our analysis/method…
  • Figure 1 and or 2. To help the reader, one or more famous town locations like Chengdu (Mianyang, Chongqing if it is inside…) can be added on the maps (especially in figure 2; 1 I think is okay as it is more geological).
  • Figure 2 Caption. Black polygon or black triangle?
  • Page 6 line 106. Double “for”
  • Page 6 middle part (lost line number). “these decomposed components…” I think that it is more correct to say that we expect that in the decomposed component it is expect to found the separated contribution of solid Earth tide, air pressure and so on…
  • Page 6 middle part (lost line number). Why the first 20 (and not 15, 17 or 22)? Please justify or insert a reference.
  • Page 6 line 120. anomaly day à anomalous day
  • Page 6 line 121. 5:12 UT right?
  • Figure 3. The ZhaoTong signal of asesimi day is more disturbed than other stations and it is strange that the noise “goes” in the “periodic components” and not in the “noise components” and this last one is less than other station (scale of 0.005 and value so much smaller). Could you check, please? And do you have control on this decomposition?
  • Page 8. Line 137. “more than 80% of…” I think is more correct “80% or more of …” if I understand well degree = 4 correspond exactly to 80% of connections of the 6 station of the network. If you consider degree equal 4 and above as anomalous than please change the sentence, otherwise if you consider >4 and not equal the text is perfect as it is already.
  • Page 8 line 141. “with higher sampling rate” is it “with higher accuracy”?… Sorry, I don’t understand the sentence.
  • Page 9 line 173. I think without “its day” is more readable.
  • 11 I would that the authors add some note on the Adj. R square that reported on Fig. 6, for example a threshold of 0.97 can be considered and most of the Sigmoidal fits pass this threshold.
  • Page 11. Line 214. “The greater the energy, the…of accumulation.” Please, revise the sentence, I think is missing a verb or something.
  • Figure 8. If you omit EQ-c than I think you need to speak in the caption about four earthquakes not 5.
  • Figure 8. EQ-a and second one about EQb and d. Please add the vertical line at the EQ time.
  • Page 12 line 266. The earthquakes cannot be located at high altitude…!!! Perhaps you want to say that the epicenters of EQa - EQd correspond to high elevation locations. In any case it is not clear for me why in this case it wouldn’t show a stress change before, please explain better/more…
  • Page 13. Line 13. A comment: I noticed a different shape of the sigmoidal fit with respect to the other ones, anyway this is partially subjective and the R-square is less in this case. What is clear is that the trend before and after the event is totally different and the slope of the 90 days before the Lushan event is so much higher than 90 days after!

Author Response

We are very grateful to your comments for the manuscript. They have important guiding significance for our manuscript and our research work. We have revised the manuscript according to your comments. The response to each revision is listed in the attachment.

Please see the attachments. The line number in the response refers to the line number in the PDF named revision.

The attachments include 3 PDF files. There are 

  1. A response to Reviewer 1
  2. A revised manuscript with highlighted changes
  3. A new manuscript

Reviewer 2 Report

In this manuscript (ms) by Zining Yu et al., a three-year (2011-2013) study for the detection of earthquake related anomalies in a borehole strain network is presented. The study is based on multi-channel singular spectrum analysis (for the identification of earthquake-related strain components) and uses a daily network reconstruction of the correlations between the measurements of six borehole strain stations in the south-western endpoint of the Longmenshan fault zone in China. A topological property of the constructed networks, which is the daily degree centrality, is investigated for the presence of anomalies. The behavior of these anomalies is studied before 13 earthquakes (Table 2) whose epicentral areas lie within the polygon defined by the six borehole strain stations or in an 100km area around them as well as 5 additional earthquakes with epicenters in a larger area. The authors identify precursory activity for 11 out of the 13 earthquakes while their results for the last 5 earthquakes show that such activity was present only for one earthquake which is the closest to the polygon defined by the six borehole stations (Fig.2). The paper analyses data collected from a large-scale experimental set-up using modern methods of statistical physics. It is well-written, and the over-all presentation is clear. As concerns the results, they are original and very promising/important in the field. For these reasons, I suggest the publication of this ms in Entropy.

Since I found only a few typing mistakes (that are listed below), I suggest publication in its present form, but the authors should take care of these typos during the proof-reading process.

Typos:

Line 85, “distance of each” -> “distance (km) of each”

Page 5, Equation (3), please define p_i and q_i (lower case) in the line before this Equation

Page 5, Equation (5), please insert (1/N) in the right-hand side of the Equation so that overbar{k} is a mean (as it is understood from Fig.4b).

Page 5, 1st line of subsection 3.2 “their raw time series for the strain sequences of” -> “the raw time series of the strain sequences for”

Page 6, Equation (8) please change subscript from omega to w.

Lines 146-147 “and found that the local earthquake energy value of 10^2 and 10^3 were the highest.” -> “and found that the most probable local earthquake energy values were 10^2 and 10^3.”

Equation (9) “x_0”-> “x0”

Line 180 “by two power-law behaviours” -> “by two different behaviours”

Line 253 “and appear self-organization” -> “and show self-organization”

Line 261 “did not deviate from” -> “did not deviate much from”

Line 497 Please fix the names of the authors in Reference [71].

Author Response

We are very grateful to your comments for our work, which is very helpful for us to improve the quality of the manuscript. We have revised the manuscript according to your comments. Please see the attachments.

Round 2

Reviewer 1 Report

Thank you very much for the good reply and to have taken into account my revision and suggestion. I’m completely satisfied with the newly obtained results and I think the paper is also stronger, as you successfully excluded a possible bias of the analysis (i.e. the acceleration due to foreshocks and not to strain accumulation). The fact that on one far EQ the strain-acceleration seems to correlate to foreshocks I think is also good, because it is reasonable that the strain method is sensible only to EQ inside/close to the network, but it is anyway sensible to foreshock of events occurring at an intermediate distance. So I confirm and reinforce the very quality of this work, as well as the proper way of analysing the data and approach to the scientific problem.

Reading again the manuscript I found some very minor point that I list down. In the reply, there is for my point of view a problem in a graph (fig. 3) but it doesn’t influence the paper.

On the reply:

The Gutenberg-Richter diagram of the reply (Fig.3) presents some “incongruences” as the cumulate number of EQs doesn’t correspond every time to the bars. For example from 6.0 to 5.5, it would be supposed to remain at the same level (0.477, i.e. 3 EQs) and only at 5.5 increment at 0.788 (i.e. 6 EQs). It looks strange too that all the M6+ are in the last bar as the Lushan EQ would be in another bar at a higher magnitude. So I don’t know exactly what I need to follow on this graph, supposing the bar from eyes inspection I think the completeness magnitude would be between 1.5 and 3. Another abnormal fact is the b-value supposed to be closer to 1.0 but it could be also 0.71, but I’m not so convinced. As this graph is only in the reply I think the paper could be anyway accepted, even I suggest to the author to revise it.

Specific points in the manuscript:

  • Page 1 line 6. “ulti-site” is it “multi-site” ?
  • Page 2 line 45. The degree symbol after 45 I think is mistyped… put this “°” instead of “â—¦”
  • Page 4 line 93-94. In the bracket you say that the hypocentral distance is measured in km and the final unit of measure is J/m2, so why is it not J/km2 or directly measured in meters?
  • Fig 2b. It would be better to change the y-axis label in log(Es (J/m2))
  • Page 11. Line 196. Please insert “is” after “the better” (the better is).
  • Page 15. Line 285. I think that for also EQ5, EQ7, EQ10 (and only for EQ6 and EQ11 Lushan) the aftershock sequence is well visible after these events.
  • Page 16. Line 315. Perhaps it is better: “discussing the earthquake precursor anomalies.”
  • Caption of Fig. 9 Perhaps it is better “scatters” instead of label in “Solid and hollow labels”
  • Page 17. Line 340. I suggest adding in the conclusion your positive result of confutation, for example adding: “that we verified that they were likely not due to foreshock activity.”

Author Response

Response to reviewer 1:

  We are very grateful to your encouragement for our work, and your specific comments are very helpful for us to improve the quality of the manuscript. We have revised the manuscript and the frequency-magnitude distribution diagram in the reply according to your comments.
